# Self-Healing and Super-Elastomeric PolyMEA-co-SMA Nanocomposites Crosslinked by Clay Platelets

**DOI:** 10.3390/gels8100657

**Published:** 2022-10-15

**Authors:** Beata Strachota, Adam Strachota, Katarzyna Byś, Ewa Pavlova, Jiří Hodan, Beata Mossety-Leszczak

**Affiliations:** 1Institute of Macromolecular Chemistry, Czech Academy of Sciences, Heyrovskeho nam. 2, CZ-162 00 Praha, Czech Republic; 2Faculty of Chemistry, Rzeszow University of Technology, al. PowstancowWarszawy 6, PL-35-959 Rzeszow, Poland

**Keywords:** nanocomposites, super-elastomers, self-healing, physical networks, self-assembly, tough elastomers, poly(methoxyethyl acrylate), montmorillonite, xerogels

## Abstract

Novel solvent-free ultra-extensible, tough, and self-healing nanocomposite elastomers were synthesized. The self-assembled materials were based on the copolymer matrix poly(methoxyethyl acrylate-co-sodium methacrylate) physically crosslinked by clay nano-platelets (‘poly[MEA-co-SMA]/clay’). Depending on the content of SMA, the super-elastomers were predominantly hydrophobic, water-swelling, or fully water-soluble, and hence repeatedly processible. The SMA co-monomer introduces a tremendous increase in tensile strength, an increase in toughness, while ultra-extensibility is preserved. By tuning the contents of nano-clay and SMA co-monomer, a very wide range of product properties was achieved, including extreme ultra-extensibility, or high stiffness combined with more moderate super-extensibility, or very different values of tensile strength. There was very attractive, great improvement in autonomous self-healing ability induced by SMA, combined with tremendously enhanced self-recovery of internal mechanical damage: even complete self-recovery could be achieved. The ionic SMA repeat units were found to assemble to multiplets, which are phase-separated in the hydrophobic polyMEA matrix. The dynamics of SMA-units-hopping between these aggregates was of key importance for the mechanical, visco-elastic, tensile, and self-healing properties. The studied super-elastomers are attractive as advanced self-healing materials in engineering, soft robotics, and in medical or implant applications.

## 1. Introduction

This article deals with the synthesis of novel ultra-stretchable tough, strong, and self-healing nanocomposite elastomers, which do not contain any solvent. The materials are of interest for engineering, robotics, and soft robotics, as well as for biomedical technologies, due to their unusual mechanical properties, their self-healing, and their bio-compatibility.

Ultra-extensible materials have attracted research interest for a long time. Extensibilities of commercially produced elastomers, such as natural or synthetic rubber, reach from 100 to 600%, and somewhat further in exceptional cases, in combination with moduli in the order of MPa [1]. Among the first ultra-stretchable elastomers, which by far surpassed the extensibility of 600%, were Haraguchi-type nanocomposite hydrogels [2]. Generally, the ultra-extensible hydrogels were easier to obtain, and were earlier ([2]) and more often described in the literature, than the solvent-free super-elastomers.

The known ultra-extensible hydrogels usually possess sophisticated architectures, especially in cases of high-modulus gels (MPa range, see [2,3]). Such architectures always contain macro-crosslinking objects (nanoparticles) of different geometry, from which a multitude of long polymer chains grows in the outward direction. These basic brush-like structure elements are interconnected to an ultra-extensible network via simple entanglements, and eventually also by some permanent crosslinks (trapped entanglements, strong adsorption of parts of chains, or covalent bonds). Organic–organic (rare) [4,5] and organic–inorganic (typical) [2,3,6,7,8,9,10,11,12,13,14,15] nanocomposite gels of this type were reported. Hyperelastic gels also were synthesized using swelling solvents other than water, e.g., ethylene glycol [8]. Record values of extensibility for gels and hydrogels were reported in the range of 12,000 to 15,000% (see [5,6], respectively), which means 120- to 150-fold elongation. A big drawback of all the above (hydro)gels, however, is their environmental instability: their properties markedly change as a consequence of drying in air, or, conversely, of water uptake (increased swelling).

Nanofillers can greatly improve the mechanical properties of a given polymer, also in the case of hydrogels. Their very high specific surface is very useful [16] for achieving strong interface interactions at small filler loadings. In cases of sufficiently small dimensions, optical transparency additionally can be preserved [6,17], while the desired chemical [18,19,20,21,22,23,24,25,26,27], optical [28,29], electrical [30,31], magnetic [32,33], or gas barrier [34,35,36] properties can be introduced into the matrix. In the super-elastomers studied in this work (as well as in the mentioned Haraguchi gels), the inorganic filler plays the role of a key structural element—poly-functional crosslink—in their sophisticated architecture. In the latter case, the heterogeneity on the nanometer scale does not reduce the extensibility, but, on the contrary, it raises it markedly (architecture effect).

Ultra-extensible solvent-free elastomers do not suffer from environmental instability, in contrast to the above-discussed (hydro)gels. However, as mentioned above, they are more difficult to obtain and are hence much less studied in the literature. This material family includes super-soft solvent-free elastomers (SSSFE) which possess similar moduli such as soft hydrogels (5000 down to 500 Pa). Such materials consist of long linear bottlebrush-type chains (grafted polymers), which are crosslinked by entanglements, or by covalent connections, as reported, e.g., in [37]. The flexible grafted sidechains play the same role which the solvent plays in (hydro)gels.

Another branch of the above-mentioned material family are solvent-free super-extensible elastomers [38], which are typically based on an architecture similar to the above-discussed Haraguchi hydrogels (polyacrylamide/nano-clay systems: [2]). In place of a solvent, however, these ultra-elastic rubbers (which also were first reported by Haraguchi) contain a higher volume percentage of elastic chains than their hydrogel analogues. These materials, based on ultra-flexible poly(2-methoxyethyl acrylate) (‘polyMEA’) matrix crosslinked by nano-clay, achieved extensibilities between 1000 and 3000% [38]. PolyMEA also was reported to be bio-compatible and hence is attractive for medical applications such as tissue engineering or implants [39,40,41]. In the present work, super-elastomers based on polyMEA nanocomposites were studied, which contained new ionic co-monomer units in the polymer chains, and thus displayed a markedly altered physical crosslinking mechanism, increased strength, and improved self-healing.

Triblock copolymers polystyrene-poly(butyl acrylate)-polystyrene [42] represent another example of rare solvent-free ultra-extensible elastomers. The morphology of these materials forms by self-assembly and nanophase separation, and leads to a morphology which, in its principle, is similar to the above-discussed polyMEA/clay system.

To sum up, from the mechanistic point of view, the combination of crosslinks between the starlike brushes of elastic chains, which are partly dynamic crosslinks (‘soft’) and partly strong ones, seems to be a key feature in all the reported advanced tough and ultra-extensible materials, both (hydro)gels and solvent-free ultra-elastomers.

In their previous work, the authors studied stimuli-responsive superporous nano-filled poly(N-isopropylacrylamide) (PNIPAm) hydrogels, which were not super-elastomeric, and in which the filler phase acted as a mechanical reinforcement, which also provided the stability of the porous structure during its ultra-fast swelling response to temperature stimuli. At first, particulate nanofillers such as nano-SiO_2_ [31,43,44,45,46] and nano-TiO_2_ [47] were incorporated. In their more recent research, the authors investigated the ultra-extensible PNIPAm/clay (Haraguchi-type) nanocomposite hydrogels and to their optimization and derivatization [48,49,50,51]. They tuned the length of elastic chains in these nanocomposites in a very wide extent (between 0.5 and 5 MDa, via initiation conditions), and in this way greatly improved their tensile properties without reducing their modulus even in case of the longest chains (entanglements helped to achieve this) [48,49]. Super-porous ultra-fast stimuli-responsive derivatives also were developed [50,51]. In most recent research [52], the authors synthesized solvent-free polyMEA/clay nanocomposites with dramatically improved tensile properties (without compromising the modulus), which was achieved by obtaining very long elastic chains. The latter elastomer, similarly to some PNIPAm/clay hydrogel varieties [11,12], indicated a potential of self-healing of disrupted samples at selected compositions [52] (which was not deeper studied), as well as considerable self-recovery of internal mechanical damage. In a subsequent work [53], the authors prepared highly transparent, tough, and ultra-extensible polyMEA/SiO_2_ nanocomposites, which exhibited very attractive elasto-plastic, or plasto-elastic behavior, as well as related polyMEA/clay/SiO_2_ systems. Another important recent work [54] was dedicated to Haraguchi-type gels, which were doped by methacrylate anion repeat units neutralized by different trivalent cations. The ionic dopant at some compositions displayed tremendous effects on the toughness and on the self-recovery behavior of the hydrogels.

In contrast to most of the above-discussed earlier works, the present one is dedicated to the rare class of solvent-free super-elastomers, and successfully demonstrates the possibility of introducing ion-related self-healing and toughening mechanisms, which were efficient in hydrogels (authors’ previous work [54]), into these new materials. The studied solvent-free polyMEA/clay super-elastomers strongly differ from the above hydrogels not only by environmental stability, but also by their high moduli (at small deformation: 1 to 10 MPa), which are typical for rubbers and stiff rubbers. The studied materials (first reported in [38]) still share a certain similarity with the Haraguchi-type hydrogels, namely the physical crosslinking of polymer chains by clay nanoplatelets, but the super-elastomers do not contain any solvent. Hence, and also because of the conditions of their synthesis (which starts in aqueous solution before yielding a dry elastomer), the studied materials could be referred-to as xerogels. In the mentioned recent work by the authors [52], the self-healing of polyMEA/clay nanocomposites with very long elastic chains was noticed, but not investigated, and elastic recovery after very large deformations was illustrated only on one example. The present work hence studies in detail the self-healing behavior of polyMEA, polyMEA/clay, and of polyMEA/clay doped with ionic groups in the polyMEA chains. The ionic groups in polyMEA were found to cause similar-type-, but much stronger effects than in hydrogels. Due to the hydrophobicity of the polyMEA elastic chains, the ionic groups additionally display a very specific micro-phase-separation behavior, which manifests itself as distinct switching of elastic behavior of the presented materials, namely between plasto-elastic, elasto-plastic, and viscous elastic type.

The aim of the present work hence was to synthesize ultra-extensible polyMEA/clay nanocomposites, whose matrix polymer would be doped by ionic sodium methacrylate units, in order to improve and fine-tune the mechanical, tensile, self-healing, and self-recovery properties. Another important goal was to elucidate structure–property relationships in this complex self-assembled nanocomposite system.

## 2. Results and Discussion

### 2.1. Synthesis of the Super-Elastomeric Nanocomposites

The studied nanocomposite super-elastomers were prepared according to Figure 1 and Table 1. The synthesis proceeded by free-radical polymerization—initiated by redox pair ammonium peroxodisulfate (APS) + tetramethyl ethylene diamine (TEMED)—of an aqueous solution of the main monomer, methoxyethyl acrylate (MEA; as polymer it is hydrophobic) and sodium methacrylate co-monomer (SMA, ionic and hydrophilic), in the presence of dispersed commercial clay nano-platelets (‘RDS’: nearly circular shape, diameter: ca. 25 nm, thickness: ca. 1 nm). The nanofiller served as polyfunctional physical crosslinker (see Figure 2 and discussion in previous works [48,49,52,53]). The ionic co-monomer SMA was added in order to modify and enhance the internal interactions in the nanocomposite elastomers, and was expected to raise their toughness. The sequence of addition of the co-initiators (see Experimental Part) led to the start of polymer chain growth on these platelets, which was of key importance for forming the crosslinking structures discussed further below.

Nanofiller content was 2, 4, and 10 wt.%, while the SMA content was 1, 5, 10, and 20 mol% of monomer units. Nanocomposites with more than 10 wt.% of the RDS clay were difficult to prepare, because the synthesis mixture becomes relatively viscous, so that increasing challenges arise, especially with homogeneous admixing of co-initiators. The employed concentrations of the latter (see Experimental Part) were based on previous experience with polyMEA systems [52,53] and were chosen so that very long polymer chains were favored, which in turn was useful for obtaining super-elastomeric properties.

Novel poly(MEA-co-SMA)/clay nanocomposites were prepared as main products onto which the research interest was focused. Simpler polyMEA/clay nanocomposites (already studied by the authors in an earlier work [52]), neat ‘linear’ polyMEA, and the ‘linear’ copolymer poly(MEA-co-SMA) with 10 mol% of SMA were prepared as reference materials.

The abbreviated sample names use ‘R’ and a number for indicating the content of RDS clay, and ‘S’ (and number) for the molar content of the SMA co-monomer. For example, ‘4R-10S’ is a nanocomposite with 4 wt.% of clay filler and 10 mol.% of SMA in the polymer matrix, ‘10S’ is a clay-free copolymer poly(MEA-co-SMA) containing 10 mol.% of SMA, while ‘4R’ is (SMA-free) polyMEA filled with 4 wt.% of clay.

Phase separation during synthesis: As described and discussed in previous works [52,53], the formation of polyMEA from the aqueous monomer solution always is accompanied by phase separation, as the monomer is moderately hydrophilic, but the polymer is predominantly hydrophobic. The micro-precipitation followed by coagulation also occurs in case of polyMEA/clay nanocomposites [52,53]. This effect leads to the appearance of turbidity and to a subsequent formation of a partly hydrated solid raw product, which has a cottage cheese consistence. The raw product is dried to yield the final elastomer sample, which then displays relatively good transparency. The described behavior is illustrated in Figure 1 bottom, and was characteristic also for most of the prepared poly(MEA-co-SMA)/clay nanocomposites, as far as the SMA content in the polymer chains was not overly high. In extreme cases, such as 20 mol% of SMA, the phase separation, as in Figure 1 bottom, did not occur altogether—in place of the cheesy raw product, a viscous solution was obtained, and for synthesis finalization, it was cast in an open mold and dried. At 10 mol% of SMA, the phase separation was distinctly reduced, but still visible and the final raw product was liquid, similarly to those with 20 mol% of SMA.

#### 2.1.1. General Properties, Hydrophobicity vs. Hydrophilicity

All the prepared dry products were elastomers, but their properties to modulus, extensibility, creep-tendency, self-healing, and self-recovery were widely varied by the composition (amount of RDS clay and SMA). The transparency was reduced at the highest clay content (10 wt.%).

The SMA co-monomer had a distinct effect which reduced the hydrophobicity of the products: The neat polyMEA and the nanocomposites with 0 or 1 mol% of SMA in the matrix displayed fairly high hydrophobicity and only a small uptake of water if immersed in a bath (see Figure 1); these samples became turbid by the absorption of the small amount of water. At 5 and 10 mol% of SMA, the nanocomposites were distinctly swelling, but not dissolving in water: they yielded opaque white hydrogels if put into a water bath. The nanocomposite with 20 mol% of SMA is fully water-soluble (and hence repeatedly processible) and yields a transparent solution, thus reversing the last step of the preparation procedure. In case of the clay-free elastomers, the ‘linear’ poly(MEA-co-SMA) elastomer with 10 mol% of SMA (‘10S’) already is water-soluble.

#### 2.1.2. Structure of Crosslinking in the Nanocomposites

In Figure 2, the basic structural features of the prepared nanocomposite elastomeric networks are shown (they were studied in detail in previous work [52,53]): Polymer chains anchored by adsorption on clay platelets (Figure 2a) are the most basic feature, which also is the basis of the distantly related ultra-extensible and tough poly(N-isopropylacrylamide) (PNIPAm)/clay hydrogels, mentioned in the Introduction. The very efficient adsorption of a long chain segment is the result of the previous adsorption of monomer, prior to its polymerization, and of starting the polymerization from equally adsorbed TEMED co-initiator molecules, as mentioned further above (Synthesis).

Characteristic for polyMEA, a polyacrylate, are covalent branching points of anchored chains (Figure 2b) formed by radical chain transfer (Figure 3). This feature, which has a great influence on the elastic properties (shape memory after large deformations) was noted and discussed by the authors in [52] for simple polyMEA/clay gels.

The entanglement of clay-anchored elastic chains (see Figure 2c, [52]) is a major source of reversible (physical) long-range crosslinking in the studied nanocomposite networks, in combination with the above-mentioned permanent branching points. The branched structure (Figure 3b) of the free segments of the clay-anchored chains (shown in simplified form in Figure 2a) contributes to making the entanglements shown in Figure 2c mechanically more efficient—in the cases where this branching does not lead to the above-mentioned permanent crosslinks shown in Figure 2b.

Finally, electrostatic interactions and nano-aggregation of the ionic SMA units (discussed further below) were found in this work to be an important crosslinking, toughening, and self-healing feature.

##### Interactions in the Nanocomposites

If the nanocomposite networks’ structure is considered at a smaller scale, the above-discussed crosslinking elements are based on the interactions illustrated in Figure 4: The polyMEA adsorption on clay nanoplatelets, which is responsible for anchoring the long polymer chains on it, can be assigned to electrostatic interactions of negatively polarized oxygen atoms of MEA repeat units with positively polarized silicon atoms in the clay structure (Figure 4a). This interaction, which leads MEA monomer to partly adsorb (prior to polymerization) on the clay was proven by ^1^H-NMR spectroscopy in [52] (sub-chapter about tuning polyMEA chain length).

Adsorption of the ionic SMA units on clay surface (Figure 4b) is a new interaction in the physical network of the novel copolymeric nanocomposites studied in the present work. The adsorption on clay can compete with the tendency of SMA to stay in aqueous solution, due to its high hydrophilicity. For elucidating the situation of SMA in more detail, a model ^1^H-NMR experiment was carried out (Figure 2). The result indicates that at the employed synthesis concentrations of SMA and RDS clay, the SMA co-monomer only partly adsorbs on clay, while 30–40% of its amount stays in solution. Full adsorption would lead to the complete disappearance of the SMA signals in Figure 2 because of slow relaxation at the given NMR experiment setup. This analysis shows that incorporation of SMA will only partly contribute to anchoring poly(MEA-co-SMA) chains on clay nanoplatelets. At the same time the SMA incorporation will raise the hydrophilicity of the free polymer chain segments (visualized in Figure 2a)—this should be expected even at low SMA contents.

Association of SMA units to ionomeric multiplets, which physically crosslink the free chain segments (see Figure 4c), is another interaction characteristic of the novel nanocomposites studied in this work. At low SMA concentration in the polymer chains, these multiplets can be highly isolated, as they are separated from each other by large hydrophobic chain segments of pure polyMEA. At higher SMA concentrations, the SMA aggregates might be fluctuating, i.e., undergoing SMA unit exchange with closely neighboring ones: this would cause smaller mechanical reinforcement, but it would still generate an energy-absorbing mechanism which would increase the toughness of the nanocomposites. A similar toughening mechanism based on ionic-aggregate disconnection/recombination was observed by the authors in [54] for hydrogels containing SMA co-monomeric units and different metal cations as counterions. The SMA association effect was indeed found to play an important role in thermo-mechanical, tensile, and self-healing properties of the studied nanocomposites, as will be discussed further below.

#### 2.1.3. Morphology (TEM)

The morphology of the synthesized nanocomposite super-elastomers was analyzed by means of transmission electron microscopy (TEM). Selected results are shown in Figure 3, while the morphology of the same samples, as well as of the highly filled 10R at different zoom is shown in Appendix A.

It can be generally observed that the incorporation of 10 mol% of SMA into the matrix leads to a more regular morphology, with more even and finer dispersion of the clay nanofiller. This ‘homogenization’ of the nanocomposites by 10 mol% of SMA can be attributed to the more hydrophilic character of the matrix polymer chains: The phase-separation at high monomer conversion (see Figure 1 bottom) becomes less dramatic with the partly hydrophilic chains, thus supporting a finer final morphology. As mentioned in the Synthesis section, a very high SMA content (20 mol%) even completely suppressed the phase-separation during synthesis and drying.

### 2.2. Thermo-Mechanical Properties (DMTA)

In Figure 4, Figure 5 and Figure 6, the analysis of the thermo-mechanical properties (DMTA) of the studied nanocomposites is summarized. The results indicate the interplay of crosslinking power of the nanofiller, ionomeric SMA–SMA interactions (which were very prominent in some cases), and of polymer-chain entanglement effects.

Contribution of the components of the nanocomposites: The Figure 4 illustrates the thermo-mechanical properties of the most attractive nanocomposite 4R-10S, and shows the contributions of its constituent components polyMEA, nano-clay crosslinker, and SMA co-monomer. In case of the neat polyMEA ‘linear polymer’, the effect of entanglement of long branched polymer chains is clearly visible (curve in Figure 4a): the polymer displays a rubbery plateau in storage modulus (G′) which is typical for crosslinked polymers, but in contrast to the latter, the modulus is decreasing with rising temperature.

Addition of clay (4 wt.%, sample 4R) increases physical crosslinking: G′ is improved by nearly one order in the rubbery plateau of the corresponding curve in Figure 4a, without significantly immobilizing the polymer: the peak of the loss factor tan(delta) namely does not shift its position in the curve in Figure 4b. ‘4R’ also is more elastic in the glass transition region, hence its tan(delta) peak is lower than in case of neat polyMEA.

Incorporation of 10 mol% of SMA into 4R yields the sample 4R-10S. This amount of SMA does not change the crosslinking density (4R and 4R-10S have nearly identical rubber moduli), but it immobilizes the polymer matrix: the step in G′ = f(T) is shifted in Figure 4a, and a broad high-temperature peak appears in the corresponding tan(delta) = f(T) curve in Figure 4b. This immobilization is attributed to SMA–SMA-interactions (ionic multiplets) between neighboring chains. This conclusion is also supported by the further-below-discussed tensile results (greatly improved toughness) and self-healing behavior (faster healing kinetics).

Effect of nano-clay loading: The Figure 5 illustrates the effect of increasing amounts of the physically crosslinking nanofiller (RDS clay) on the thermo-mechanical properties of poly(MEA-co-SMA)/clay nanocomposites (all with 10 mol% of SMA). The clay generally increases the modulus in the rubbery region as would be expected, albeit in a non-linear way. It also raises the glass transition temperature, which also would be expected, but, as will be explained below, the main origin of this effect was found to be the SMA–SMA interactions. As reference material, the SMA-free polyMEA/clay nanocomposites were also characterized (see Appendix A; similar systems were also studied by the authors in [52]).

In the simple polyMEA/clay systems (similar trends to in Figure 5, but somewhat different: see Appendix A), the crosslinking density and hence the modulus at first slightly decreases if comparing neat polyMEA with the sample 2R loaded with 2 wt.% of clay. This can be attributed to a less efficiently entangling arrangement of polymer chains (‘disorder effect’) in the nanocomposite structure (Figure 2), which is more complex than that of entangled neat polyMEA. If going from 2R to 4R, the rubbery modulus increases by ca. half an order, and moderately surpasses neat polyMEA (‘disorder effect’ + more physical crosslinker). At 10 wt.% of nanofiller (sample 10R) the rubbery modulus dramatically increases in comparison to 4R, by ca. 2 orders. This seeming ‘percolation effect’ appears to be purely caused by physical crosslinking, via entanglements of free polymer chain ends (crosslink structure in Figure 2c further above), because the glass transition temperature (T_g_) does not change in the samples’ series 2R, 4R, 10R (see Appendix A). If the ‘percolation’ were caused by polymer adsorption and immobilization, T_g_ would rise distinctly. Moreover, true mechanical percolation of the nanofiller platelets would most likely greatly reduce the extensibility, which was not observed; however (see further-below discussed tensile properties), 10R exhibits a similarly high ultra-extensibility to 2R and 4R.

Generally, in all the SMA-free nanocomposites, the value of the glass transition temperature (T_g_) is slightly lowered by clay, relatively to neat polyMEA (−23 °C), but T_g_ practically does not change between filler loadings of 2 and 10 wt.% (−26 to −29 °C). The slight downshift of T_g_ was attributed to the mentioned ‘disorder effect’.

In the poly(MEA-co-SMA) nanocomposites, which contain the ionic SMA units, the effect of increasing amounts of clay is similar to neat polyMEA matrix, but the modulus trend is simpler (see Figure 5a): In the series 10S (clay-free poly[MEA-co-10mol%SMA]), 2R-10S, 4R-10S, and 10R-10S, the rubbery modulus increases in a simple trend. The great increase between 4R-10S and 10R-10S (‘percolation’) is somewhat less dramatic than in the neat polyMEA matrix. In analogy to the SMA-free specimens, the trend can be assigned to the increasing amount of the crosslinking nano-filler. The ‘disorder effect’ on rubber modulus in case of 2R-10S is somewhat hidden, as the sample has a slightly higher modulus than neat polyMEA, and a markedly higher modulus than the ‘proper reference’ 10S (poly[MEA-co-10mol%SMA]). With exception of the sample 10R-10S, the rubbery moduli of the SMA-doped samples are always higher, than the moduli of their SMA-free analogues.

The glass transition behaviour of the poly(MEA-co-SMA)/clay nanocomposites displays an interesting new trend: In contrast to the SMA-free nanocomposites, the glass transition region of the SMA-doped ones (see Figure 5) depends on clay content, shifts to higher temperatures and becomes broader with increasing amounts of nano-clay. By analyzing the tan(delta) = f(T) curves in Figure 5b it can be observed, that all the nanocomposites display a rather small peak which is assigned to the glass transition of the freely mobile polymer chain segments. This peak is downward-shifted (to −30 °C; 10R-10S: −40 °C) in comparison with neat polyMEA (‘disorder effect’), somewhat more distinctly than in polyMEA/clay. The small ‘peak of the free chains’ is adjacent to a wide and very broad peak at higher temperatures (ranging from −30 up to ca. +70 °C, with maximum between 0 and +20 °C), which corresponds to the glass transition of immobilized polymer chains. Comparison with the clay-free sample 10S (see curve in Figure 5b and detail view in Appendix A), which contains only this broader peak without the smaller ‘free-chains-peak’ at low T, indicates that the additional immobilization in the poly(MEA-co-SMA)/clay nanocomposites is caused mainly by SMA–SMA interactions and not exclusively by SMA–clay interactions. The latter would be the first choice as explanation, but 10S with all chains free and no adsorption on filler only displays the broad high-temperature-peak. The SMA–SMA interactions will be shown to play a key role also in tensile and self-healing properties.

In Figure 6 the effect of increasing SMA co-monomer percentage (1–20 mol%) in the polymer chains of the nanocomposite matrix is analyzed in detail on the example of the mechanical properties of the nanocomposites containing 4 wt.% of RDS clay. The SMA co-monomer displays a strong effect on DMTA profiles, with relatively complicated trends. The effect of SMA generally can be attributed to the formation of the already mentioned ionic multiplets depicted further above in Figure 4c, which are more or less phase-separated in the volume of the matrix copolymer, depending on the ratio of MEA and SMA repeat units.

Glass transition behavior: In the range between 0 and 10 mol% of SMA, the value of T_g_ steadily increases with SMA content: this is well visible as shift of the step in G′ in Figure 6a. At 10 mol% of SMA (sample 4R-10S), the immobilization of the chains is the strongest. However, in Figure 6b (tan(delta) = f(T) curves), it also can be observed that even at 10 mol% of SMA, a small fraction of the chains still is not immobilized, and displays the low-temperature glass transition (peak at −30 °C).

At 20 mol% of SMA (sample 4R-20S) the immobilization effect recedes; however, this is most likely due to the more homogeneous character of the polymer and hence to easier hopping of SMA units between neighboring ionic domains (nano-phase-separation of SMA is no more efficient). The corresponding step in the modulus (G′) for the sample 4R-20S recedes approximately to the same temperature as is found in case of 4R-5S.

The rubbery modulus is most strongly increased at very low concentrations of SMA in the polymer matrix: 1 mol% (lowest tested content) raises the rubbery modulus by nearly one order in comparison to the SMA-free sample 4R. At 5 mol% of SMA, the rubbery modulus is less strongly increased, at 10 mol% it is practically the same as in 4R, and at 20 mol% of SMA, the modulus is smaller (4R-20S in Figure 6a) than that of neat polyMEA. The strong crosslinking effect of SMA co-monomer at low concentrations can be assigned to hydrophobic repulsion between multiplets of SMA units (Figure 4) and chain segments of pure polyMEA. The spatial isolation of the multiplets is most efficient at low SMA contents. At higher SMA percentage, the hopping of SMA units between not-so-distant SMA domains is easier. Additionally, at 20 mol% of SMA, it seems that intra-chain interactions SMA–SMA are preferred over inter-chain ones, thus leading to less efficient entanglement and physical crosslinking (according to structure in Figure 2). As was noted further above, the aqueous synthesis mixture 4R-20S does not precipitate or gelate (and stays transparent) prior to the moment where the product is dried to obtain the final solid specimen. Generally, 20 mol% of SMA reduced the mechanical as well as the tensile properties (as will be discussed further below).

The SMA effect is similar in nanocomposites with different clay loadings, as is demonstrated in Appendix A, where nanocomposites with clay loadings from 2 to 10 wt.% are compared with their analogues which contained 10 mol% of SMA in the polyMEA matrix. Some differences can be seen; however, the up-shift of T_g_ is more pronounced at higher clay contents. At 10 mol% of SMA, the rubbery modulus was slightly increased if 2R and 2R-10S are compared. It is unchanged in case of 4R vs. 4R-10S, and it is distinctly decreased in case of 10R vs. 10R-10S. In the latter case, intra-chain SMA–SMA interactions might contribute to the observed behavior, and might be favored by a higher volume fraction of hydrophilic clay on which the polymer chains are anchored. Indeed, the relatively modest self-healing properties of 10R-10S (discussed further below)—in comparison with other SMA-doped nanocomposites—support the assumption about increased intra-chain interactions in this material.

### 2.3. Phase Transition Behaviour and Thermal Properties (DSC)

The thermal properties of the prepared nanocomposites, and especially their glass transitions, which manifest themselves as step-like changes in heat capacity, were investigated by means of differential scanning calorimetry (DSC). Exemplary DSC traces are shown in Figure 7 (more traces are shown in Appendix A), while the obtained results are summarized in Table 2 (see Section 4.4.4). It can be noted, that all the analyzed samples (polyMEA, polyMEA/clay-, and poly(MEA-co-SMA)/clay nanocomposites) display glass transitions well-visible in DSC. These are very distinct in case of neat polyMEA or of nanocomposites based on SMA-free polyMEA matrix (example: Figure 7a), as relatively steep steps in heat capacity. This kind of glass transition seems to be characteristic for polyMEA chains.

Clay content: In SMA-free nanocomposites, increasing loading with clay reduces the height of the step in heat capacity at T_g_; however, the step always stays fairly steep, even with 10 wt.% of clay. This effect is not surprising, because the mass of the inorganic clay does not take part in the glass transition process itself and in the associated changes in heat capacity.

In case of SMA-doped nanocomposites (example: Figure 7b) there are typically two adjacent smaller steps during the first scan, and only the lower-temperature step visibly reappears during the second scan. The ‘metastable’ higher-temperature step was assigned to the mentioned SMA–SMA interactions, which also influence glass transition behavior in the further-above discussed thermo-mechanical tests. In the second scan, the higher-temperature glass transition step is practically invisible: it is not steep anymore, and it is spread over a wide temperature range; however, the results in Table 2 show that the change in heat capacity for the flat (‘hidden’) transition is approximately the same as in case of the first scan. The highly filled and SMA-doped 10R-10S sample (DSC: see Appendix A) displays only one broader step in heat capacity already in the first scan, which is in the same temperature range, as in the glass transition features in the corresponding tan(delta) = f(T) curve in Figure 5b. In the latter thermo-mechanical analysis (in contrast to DSC) the small and strongly down-shifted peak (at −40 °C) generated by ‘polyMEA’ free chains still can be discerned as a feature overlaid over the broad higher-temperature transition.

The T_g_ values observed by DSC (see Table 2) did not significantly change with the samples’ composition: neat polyMEA displayed T_g_ = −30.8 °C, whereas the ‘polyMEA step’ in the nanocomposite samples, both SMA-free and SMA-doped, was always between −31.2 and −32.7 °C. An exception was the highly filled 10R-10S, where the ‘polyMEA’ step merged with the ‘high-temperature-step’ of the polymer chains immobilized by SMA–SMA interactions (T_g_, = −15.6 °C).

The presence of a certain amount of residual water also was detected by DSC (see Figure 7 and Appendix A): All samples (neat polyMEA, polyMEA/clay- and poly(MEA-co-SMA)/clay nanocomposites) display a very broad endotherm in the first scan, which completely disappears in the second. This feature was assigned to the evaporation of residual water. Presence of clay nanofiller, or of the hydrophilic SMA co-monomer, generally leads to a larger ‘evaporation endotherm’. As was mentioned in the section Synthesis, all the materials, including neat polyMEA are able to take-up some water via swelling. Small amounts of residual water also can be helpful in the dissociation and recombination of SMA–SMA multiplets, a mechanism which plays an important role in the mechanical, tensile, and self-healing properties, as will be discussed further below.

### 2.4. Tensile Properties: Simple Tests

The studied polyMEA-based materials were all highly extensible or even ultra-extensible. The results of tensile tests are summarized in Figure 8. The extensibility of the studied materials was mainly of elastic nature, as will be discussed further below (cyclic tensile tests): very large tensile deformations are retracted to a large degree even in case of materials with ‘plastic-like’ tensile curves, e.g., 10R or 4R-5S from Figure 8. All the simple tensile tests were performed with thick ‘laminated’ specimens (such specimens were also used for the self-healing tests of samples which were cut and re-connected; their preparation exploited the self-healing property: see Experimental Part).

Clay loading effect: The effect of increasing clay loading on the tensile properties of the simple reference system polyMEA/clay is shown in Figure 8a. Incorporation of the RDS clay as macro-crosslinker leads to more than two-fold increase in extensibility, which was similar at all the tested loadings (2–10 wt.% of clay). The tensile toughness markedly increased with clay content, especially between 2R and 4R, and extremely between 4R and 10R. The stress at break similarly increased in the last two cases. The trends can be attributed to the complex structure of the nanocomposites (Figure 2), which offers additional strong and weak physical crosslinks, as well as additional energy absorption mechanisms.

SMA concentration effect: The effect of increasing SMA concentration on the tensile properties of poly(MEA-co-SMA)/clay nanocomposite elastomers with a constant loading of 4 wt.% of clay is illustrated in Figure 8b. In general, it can be observed that increasing SMA content dramatically increases the toughness and tensile strength, as well as the differential Young’s modulus at large deformations, while it slightly reduces extensibility. An exceptional sample is 4R-5S with a relatively low SMA content, which reproducibly displayed very high ultra-extensibility (elongation at break: ca. 3400%). As was noted further above, the samples with low SMA content, 4R-1S and 4R-5S were found to display ‘anomalous’ mechanical properties (DMTA), which were attributed to efficient nano-segregation of ionomeric multiplets of SMA. In case of 4R-5S, the dissociation and recombination of these multiplets is easier than in 4R-1S—it generates an energy-absorption mechanism but does not generate crack defects, so that ultra-high extensibility is supported. In case of 4R-10S, the numerous SMA–SMA interactions contribute to a very high tensile strength, but due to easier SMA hopping between aggregates, the elongation at break is no more ultra-high, but approximately the same as in the case of the SMA-free nanocomposite 4R. The sample 4R-20S preserves the ‘standard’ extensibility of 4R, but in comparison to the less ionic 4R-10S, it displays a distinctly lower toughness improvement, and distinctly lower tensile strength. This finding was assigned to the very easy SMA-hopping in 4R-20S, as well as to more strongly preferred intra-chain SMA–SMA interactions, as discussed further above in the section about thermo-mechanical properties (DMTA).

Clay loading effect in poly(MEA-co-SMA)/clay nanocomposites: The effect of increasing clay loading in nanocomposites doped with 10 mol% of SMA in the matrix is illustrated in Figure 8c. The trends are somewhat similar to the SMA-free nanocomposites, but more complicated, while the curves have a completely different shape: elastomer-like (SMA-doped) rather than plastic-like (SMA-free). Incorporation of 10 mol% of SMA into ‘linear’ clay-free polyMEA leads to a tremendous increase in extensibility (to ca. 1700%), toughness and tensile strength. This is the effect of SMA–SMA interactions alone. In case of the nanocomposites, the tensile strength greatly increases and subsequently somewhat decreases with clay content, but always stays much higher than in the case of 10S. The toughness initially surpasses (2R-10S) the one of 10S, thereafter it continuously drops. Elongation at break somewhat decreases with clay content: 2R-10S is still ultra-extensible with 1200%, while 4R-10S and 10R-10S are super-elastomeric with extensibilities of 800–1000%. In case of the highest clay content (10R-10S), the yield point is at the highest stress value among the compared products (ca. 0.65 MPa), but the yield stress and the long-range differential Young’s modulus are smaller than in 4R-10S. This latter result is indicative of less efficient entanglements at a lower ratio polymer/physical macro-crosslinker. Generally, the results illustrate an interplay of SMA–SMA interactions and physical crosslinking effects by polymer/clay supramolecular assembly and subsequent entanglement.

### 2.5. Self-Healing of Disrupted Samples and Its Efficiency

Self-healing was tested on ‘laminated’ thick samples (preparation: Experimental Part). Each of them was cut in two pieces, which were subsequently put together, and manually pressed together for 30 s by a pressure of ca. 220 MPa (see Figure 9). Thereafter, the self-healing was left to proceed without any pressure or other stimulus at room temperature, for different periods of time (1h, 24h, 3 days, 7 days). Finally, tensile tests were performed with the samples after the specific healing times, and compared to tensile results obtained on intact ‘laminated’ thick specimens of the same materials. Figure 10 (neat polyMEA and 10S); Figure 11, Appendix A summarize the self-healing results. The results indicate a marked autonomous self-healing ability of all the studied polyMEA-based materials if subjected to the above simple procedure: all displayed distinct, albeit not complete self-healing. The SMA co-monomer was found to markedly improve the self-healing ability of the nanocomposites, which was attributed to the dynamics of the SMA nano-aggregates.

Under the simple self-healing conditions, neat polyMEA displays the second fastest and second most efficient autonomous self-healing (Figure 10a): 65% of the initial ‘intact’ tensile curve are restored after 7 days, most marked is the healing after 24h. Incorporation of 10 mol% of SMA (sample 10S) tremendously improves not only the ‘intact’ tensile curve (discussed further above), but also the self-healing ability and kinetics (see Figure 10b): 84% of the tensile curve is restored after growing-together for 7 days, 75% after 1 day, and 56% after 1 h. However, neat polyMEA is a very soft material with poor processing properties and also with the markedly smallest extensibility among the studied products. The clay-free ‘10S’, in spite of many interesting properties, also is a problematic sticky material. The comparison of these two related reference materials illustrates the enhancement of the self-healing property by the ionic SMA co-monomer.

In Figure 11, the results of self-healing tests are shown for the nanocomposite materials 4R and 4R-10S (Figure 11a and Figure 11b, respectively), as well as for the highly filled 10R and 10R-10S (Figure 11c and Figure 11d, respectively). In both sample pairs, always the tougher, SMA-doped material displays a visibly more pronounced self-healing: In case of 4R vs. 4R-10S, 32% of the ‘intact’ tensile curve is recovered in 7 days without SMA (4R), vs. 53% in 7 days with SMA co-monomer (4R-10S).

The highly filled systems 10R and 10R-10S (Figure 11c and Figure 11d, respectively) demonstrate an analogous regeneration-enhancing effect of SMA, but their self-healing ability is markedly smaller than in 4R and 4R-10S: In case of 10R, 13% of the tensile curve is restored in 7 d, vs. 27% for 10R-10S after the same healing time. It appears that the reduced volume fraction of polymer and the higher stiffness (high rubbery modulus at room temperature observed by DMTA) of the samples filled with 10 wt.% of clay play a key role in their slow self-healing.

The low-filled systems 2R and 2R-10S (Appendix A, respectively) also demonstrate an analogous trend, as in 4R vs. 4R-10S. However, also in the case of the low-filled nanocomposites, the regeneration of the cut samples is slower than in case of 4R vs. 4R-10S, albeit the difference is not dramatic: 2R regenerates to 20% of the intact curve in 7 d, 2R-10S to 39% in the same healing period. This result also indicates that the crosslinking density generated by the clay nanoplatelets enhances the self-healing process (as far as the filler content is not too high).

The low-doped nanocomposite 4R-1S (Appendix A) also displays self-healing, but is less efficient than most of the other studied nanocomposites: it regenerates only 24% of its ‘intact’ tensile curve in 1 week, most of which occurs in 1 d. This sample is ultra-extensible, but it also displays marked plasticity in addition to elasticity. The low content of SMA does not seem to support the dynamic reorganization of physical crosslinking via SMA hopping from aggregate to aggregate. This nano-separation of SMA was also noted by DMTA analysis further above.

Short-time self-healing (1 h): A moderate rubbery extensibility, ca. 100–150% on the absolute strain scale, is restored in all the tested materials already after 1h. This, however, corresponds to different % of regeneration of the intact tensile curve, depending on material. This rapid initial self-healing might be an intrinsic property of polyMEA, attributed to its conformational dynamics.

### 2.6. Elasticity vs. Creep after Large Stretching Deformation: Cyclic Loading Tests

The studied elastomers and super-elastomers all displayed a high or even very high elastic character also at large deformations, as mentioned in comments to the tensile and self-healing tests. The relative elasticity vs. creep was evaluated in cyclic tensile experiments, the results of which are shown in Figure 12 (extreme cases) and in Appendix A. All the tested polyMEA-based samples display marked immediate elastic retraction, but also a significant short-term creep. In general, the amount of clay and the fraction of the SMA co-monomer were found to affect the relative elasticity in a similar way, such as how both parameters affected the self-healing ability. The cyclic tests were performed with thin (‘film-like’) specimens, the thickness of which was 0.3–0.4 mm, which were more easily and quickly obtained than thick (‘laminated’) ones. The ‘film’ samples generally displayed a higher elongation at break than the thick ones, which were used for the basic-tensile-, and for the self-healing tests. The cyclic tests were performed with maximum elongation set approximately to 50% of the average elongation at break of the respective material in the form of ‘film’ (not the thick or laminated sample).

The Figure 12 shows typical examples of mainly elastic (4R-10S: Figure 12a) and of elasto-plastic (10R: Figure 12b) nanocomposites. The former shows a fairly high retraction of the maximum deformation endured in the initial cycle in the moment when the second cycle begins. The latter example shows a more plastic sample with a significant creep persisting at the start of the follow-on cycle. For the subsequent cycles, the curves appear flatter because of this initial creep.

If the effect of clay content and of absence vs. presence of 10 mol% of SMA co-monomer is systematically evaluated (Appendix A), it can be observed that the immediate creep at the end of the first cycle in SMA-free samples amounts to 30–40% (increasing with clay content) of the initial elongation value, and to ca. 25% in the samples doped with 10 mol% of SMA. The immediate creep always is visibly reduced by further slow sample retraction before the follow-on cycle starts. Finally, the cyclic curves of each specimen stabilize after a few cycles, become somewhat ‘flatter’, and the hysteresis becomes smaller. The initial cycle always displays a great hysteresis. At higher clay contents, the ‘flattening’ of curves which is caused by creep becomes more distinct. Doping with 10 mol% of SMA always visibly reduces the immediate creep, as well as the ‘flattening’ effect. Both effects are most distinct at the highest clay loading of 10 wt.%.

The effect of the fraction of SMA in the polymer matrix is evaluated in Figure 13, for nanocomposites with 4 wt.% of clay loading. If going from 4R to 4R-1S and 4R-5S, a shift is observed from relatively elastic to marked plasto-elastic behavior with considerable creep, as well as a very distinct flattening of the curves in the first follow-on and the subsequent cycles. The sample 4R-1S is more plastic (creeping) at high deformations: it does not survive the second follow-on cycle. The sample 4R-5S is much more stable and less creeping, but the hysteresis of the follow-on cycles is extremely slim, and their curves are very flat. Nevertheless, this sample is ultra-extensible and retracts a dominant part of the enormous maximum cyclic deformation of 2500%. At higher SMA contents (4R-10S and 4R-20S), the elastomer-like character is abruptly restored and improving. The anomalous behavior of the low-doped samples 4R-1S and 4R-5S was noted with several characterizations, and assigned to the effect of nano-phase-separation of ionic multiplets of SMA units, where hopping of SMA units between aggregates is difficult, thus leading to strong SMA–SMA physical crosslinking, but also to plasticity at large deformations. At higher SMA contents, SMA hopping is no more so hindered, and the SMA–SMA interactions just generate additional toughness, and eventually they also reduce crosslinking by polymer-polymer entanglements in cases of very high SMA contents, such as in 4R-20S (see Section 2.2). If going from 4R-1S to 4R-20S, the increasing SMA content gradually reduces the value of immediate creep.

### 2.7. Internal Self-Healing: Differently Fast Self-Recovery of Mechanical Properties after Large Stretching Deformation

Self-recovery tests (sometimes also referred-to as ‘internal self-healing’) are performed to evaluate the healing of internal damage done to crosslinking in a material during its large deformation. In case of the studied nanocomposites, this test characterizes the results of such internal self-healing, combined with slow retraction of a seemingly plastic deformation.

The Figure 14 illustrates the self-recovery behavior of selected nanocomposites, while the Appendix A shows an overview of the remaining samples with increasing clay content, which were SMA-free, or doped with 10 mol% of SMA. In contrast to above-discussed continuously repeated cyclic tests, the self-recovery tests (which have a similar graph appearance) were performed using a wholly different procedure: three specimens of each material were needed to complete the self-recovery test. At first, each of these specimens was subjected to a standard cyclic deformation (label ‘intact’) up to ca. 50% of its (previously known) average elongation at break. After this first cycle, one follow-on cycle was started after a pre-defined delay (time of self-recovery), which was either 0 s (label ‘immediate’), or 30 min, or 60 min. Each of the specimens hence endured only the initial (‘intact’) and one follow-on cycle (‘immediate’, or ‘30 min’, or 60 min’). The curves for one of the ‘intact’ cycles, and for the three follow-on cycles recorded after the respective recovery time periods, were then put into one graph characterizing a given material. With increasing self-recovery time, the shape, size, and position of the follow-on cyclic curves approach the initial (‘intact’) one.

In Figure 14 and Appendix A it can be noted, that the presence of 10 mol% of SMA combined with increasing nano-clay loading leads to increasingly complete self-recovery, which is nearly perfect in case of 4R-10S and 10R-10S. On the other hand, increasing clay content in absence of SMA (see Figure 14 and Appendix A) leads to deterioration of self-recovery, especially in 10R—the latter practically does not self-recover even after 60 min.

Mechanism: It can be concluded that the dynamics of the SMA-nanoaggregates greatly improves the self-recovery, and that increasing clay content (in presence of SMA) supports it, possibly via the formation of permanent elastic crosslinks shown in Figure 2b.

SMA concentration effect: The effect on self-recovery of the SMA content in the polymer matrix is evaluated in Figure 15, for nanocomposites with 4 wt.% of clay loading. Similarly, as in all the remaining mechanical tests, the nanocomposites with low contents of SMA (1 and 5 mol%) display ‘anomalous’ plasto-elastic behavior (in contrast to more elastic and self-recovering SMA-free sample ‘4R’). 4R-1S and 4R-5S are characterized by considerable creep and only a small self-recovery (4R-1S reproducibly fails if 30 min of recovery time is tested). Increasing SMA percentage to 10 mol% abruptly improves self-recovery, which becomes nearly complete, so that 4R-10S can be considered to be a nearly perfectly recovering super-elastomer, albeit viscous (with slow retraction). At 20 mol% of SMA, the self-recovery somewhat worsens again: although the retraction of initial deformation is complete already after 30 min, the mechanical properties (shape and size of the cyclic curve) are only partly recovered even after 60 min, thus indicating persisting ‘internal damage’.

Mechanism details: Both the plastic behavior of 4R-1S and 4R-5S, as well as the limited recovery in 4R-20S, seem to be connected with the earlier-discussed dynamics of nano-aggregates of SMA units: In the first two cases, strongly isolated SMA multiplets generate crosslinks which disconnect irreversibly and thus favor plastic deformation. In case of 4R-20S, easy hopping of SMA units between aggregates, as well as a tendency to intra-chain SMA–SMA interactions seem to be at the origin of reduced mechanical properties after full retraction of deformation (via re-organization of the structure of SMA–SMA crosslinks). Only the dynamics of ionic aggregates at 10 mol% of SMA yields self-recovery which is optimal in all aspects.

## 3. Conclusions

-Novel solvent-free nanocomposite super-elastomers based on the copolymer matrix poly(methoxyethyl acrylate-co-sodium methacrylate) physically crosslinked by clay nano-platelets (‘poly[MEA-co-SMA]/clay’) were synthesized;-Depending on the SMA content, the super-elastomers were predominantly hydrophobic, water-swelling but stable against dissolution, or fully water-soluble and hence solution-processible;-Generally, in the dry state the SMA co-monomer introduces a tremendous increase in tensile strength, combined with an increase in toughness, while ultra-extensibility similar to the simpler ‘poly[MEA]/clay’ nanocomposites is preserved;-Variation of composition parameters makes possible to obtain a very wide range of product properties, including extreme ultra-extensibility, or high stiffness in rubbery state combined with more moderate super-extensibility, or very different values of tensile strength;-The SMA co-monomer introduces a great improvement in self-healing ability of specimens which were cut, in comparison to the simpler poly[MEA]/clay systems; the best results were achieved with 10 mol% of SMA in the matrix and 4 wt.% of nano-clay filler (macro-crosslinker); the regeneration was 53% of the original tensile curve after 1 week at simple conditions, in case of the mentioned most attractive product; variation of SMA content switches the elastic behavior at large deformations between plasto-elastic, elasto-plastic, and viscous elastic type;-Self-recovery of mechanical damage after large deformations was tremendously improved by the SMA co-monomer, and even complete self-recovery was achieved; the best results were obtained with 10 mol% of SMA and not-too-low clay content;-Elucidation of structure–property relationships explained the principle of the tremendous effect of the SMA co-monomer on mechanical and self-healing properties: it is based on the formation of multiplets (nano-aggregates) of the ionic SMA units, which are phase-separated in the hydrophobic polyMEA matrix: at low SMA contents, the multiplets can form strong, highly isolated, and irreversibly dissociating physical crosslinks which support more plasticity; or, at higher SMA contents, the multiplets can dynamically exchange SMA units, because they are less separated, which in turn supports rapid self-healing;-The studied super-elastomers are attractive for potential applications as advanced self-healing materials for engineering, robotics, medical or implant technology.

## 4. Materials and Methods

### 4.1. Materials

2-Methoxyethyl acrylate, the main monomer, (abbreviation: MEA; Product Nr.: M2282, reagent grade, purity > 98%) was obtained from Tokyo Chemical Industry Co., Ltd. (short name TCI, Tokyo, Japan) and used as received without further purification. Sodium methacrylate (abbreviation: SMA; Product Nr.: 408212, reagent grade, purity 99%), N,N,N′,N′-tetramethylethylenediamine (abbreviation: TEMED; Product Nr.: 411019, grade: ‘purified by redistillation’ by the manufacturer, purity ≥ 99.5%), and ammonium persulfate (abbreviation: APS; Product Nr.: 215589, reagent grade, purity 98%) were purchased from Sigma-Aldrich (Burlington, MA, USA) and used as received without further purification.

The synthetic hectorite clay, ‘Laponite RDS’ (chemical composition: Na_0_._7_[(Si_8_Mg_5_._5_Li_0_._3_)O_20_(OH)_4_]; Product code: ‘Laponite RDS’, laboratory grade, purity 90%: remaining 10% = adsorbed water), which consisted of approximately circular platelets (diameter ~ 30 nm, thickness ~ 1 nm) and which was modified with pyrophosphate ions (P_2_O_7_)^4−^ (as dispersion-enhancing agent), was friendly donated by BYK Additives & Instruments (Wesel, Germany). The 10% water content present in ‘RDS’ was taken into account in the calculation of the amount of this clay, which was needed for a given synthesis.

### 4.2. Synthesis of the Nanocomposite Elastomers

The nanocomposite elastomers poly(MEA-co-SMA)/clay were prepared by in-situ free-radical polymerization of MEA and of the ‘dopant’ SMA co-monomer in water, carried out in presence of clay nanoplatelets which acted as physical crosslinker and filler (Laponite RDS), and which were dispersed in water (synthesis solvent) prior to the synthesis. The neat polyMEA matrix and the clay-free copolymer poly(MEA-co-10mol%SMA) (with 10 mol% of SMA co-monomer) were also prepared as a reference materials. After completed polymerization, the products were dried in order to obtain the final solvent-free elastomer form.

Amounts of components used to prepare the studied materials are listed in Table 1.

The following parameters were kept constant: the concentration of the polymerizable C=C bonds (MEA monomer and SMA co-monomer, both mono-functional) in the reaction mixture was 0.75 mol/L, the molar ratio of *n*(APS)/*n*(C=C) = 0.00435, and the ratio of *n*(TEMED)/*n*(C=C) = 0.0148. The molar fraction of the the SMA co-monomer was varied in the range: 0 (reference samples), 5, 10, and 20 mol% of polymerizable double bonds. Clay loading was 0 (reference samples), 2, 4, and 10 wt.% in dry nanocomposite. The abbreviated sample names are listed in Table 1: ‘R’ symbolizes the clay, and ‘S’ the sodium methacrylate (SMA) co-monomer: e.g., ‘4R-5S’ is a nanocomposite with 4 wt.% of clay filler and 5 mol.% of SMA in the polymer matrix.

The synthesis procedure started by preparing a homogenous aqueous dispersion of RDS, which was obtained by intensively stirring (magnetic stirrer, 800 rpm) RDS in water for 24 h (this exfoliation process was studied in detail in a previous work of the authors [48]). The amounts of RDS clay and of water used for each sample are listed in Table 1; the stirring was carried out in air atmosphere in a 100 mL glass vial.

In the next step, the MEA monomer and the SMA co-monomer were added (amounts: see Table 1) to the previously prepared RDS dispersion, and the solution was purged with argon under continued stirring for 5 min (the argon gas was introduced into the reaction mixture using a glass capillary). Next, one of the redox co-initiators, TEMED, was added and the mixture was further stirred and purged with argon for one more minute. Finally, the second co-initiator APS (as a 1% aqueous solution) also was added. After a brief final stirring (30 s, continued argon flushing), the reaction mixture was transferred (using a 50 mL syringe with a needle of 0.75 mm internal diameter) into an argon-filled transparent mold (internal dimensions: see detailed description below), which consisted of two removable glass sheets enclosing a rubber spacer. After the transfer, the empty space over the reaction mixture in the mold was flooded with argon for 20 s. The reaction was left to run in the mold at 25 °C for 24 h. The resulting product was usually a white opaque hydrogel, except some products with very high SMA contents which were clear or turbid viscous liquids (these are marked with an asterisk in Table 1). The latter samples were (except the very first syntheses) usually prepared in the glass vials in which the components were mixed, and were not transferred into the glass mold like the remaining specimens.

To finalize the synthesis, in case of the solid products, the hydrated and soft raw product was taken out of the mold via its disassembling (one glass plate was removed, thereafter the rubber spacer, thus leaving the obtained solid on the second glass plate). It was placed onto a Teflon plate and dried at room temperature for 24 h, and finally for 24 h at 50 °C under vacuum, in order to obtain the solvent-free elastomer.

In case of the viscous liquid (SMA-rich) raw products, these were carefully poured from the reaction vial into an open circular Teflon mold (geometry: see detailed description below) and they were dried at room temperature for 24 h, and finally for 24 h at 50 °C under vacuum, in order to obtain the respective solvent-free elastomer.

Geometries of employed molds: For the products which were formed as solids in the raw form, the following geometries of the rubber molds (between glass plates) were used: 10 × 5 × 2 cm^3^ for obtaining thick samples (volume: 100 mL; half-filled during synthesis), and 10 × 5 × 1 cm^3^ for obtaining thin samples, from which ‘films’ were cut (volume: 50 mL). For the products obtained as liquids in the raw state, an open circular mold was employed (diameter: 4 cm, depth: 0.5 cm) for the post-synthesis drying, which was made from a Teflon plate (volume: 25 mL; each synthesis yielded material for two such molds). After drying, circular samples were obtained, most of whose area had ‘film’ thickness.

Preparation of thick (‘laminated’) samples:

For the purpose of self-healing tests and of reference tests with intact samples, ‘thick’ samples were prepared either directly (thick rectangular platelets), or from the primarily synthesized circular ‘film samples’ (in case of the SMA-rich products). For this purpose, fifteen films (circular-shaped) of a given material were put over each other, and pressed between two Teflon plates by the pressure of 3 kPa, for the duration of 30 days, at room temperature. The self-healing properties of the materials yielded monolithic compact and thick circular platelets at the end of this procedure with a thickness of ca. 4.5 mm.

### 4.3. Self-Healing Tests

The samples for self-healing were obtained by cutting into two pieces ‘laminated’ thick specimens (preparation is described above), which were of platelet shape (original specimen size: same as for the below tensile test: length: 30 mm, width: 5 mm, thickness: 4.5 mm). These two pieces were subsequently ‘re-assembled’ and pressed together by a force of ca. 5000 N (corresponding to a pressure of 220 MPa at the given cross-section), at room temperature (25 °C), for the duration of 60 s. Finally, the self-healing sample was subjected to a specific ‘healing time’ during which the regeneration occurred without any pressure at room temperature. The ‘healing time’ values were typically: 1 h, 1 day, and 7 days (1 week).

### 4.4. Characterization

#### 4.4.1. NMR

^1^H NMR spectra were recorded on an upgraded Bruker (Karlsruhe, Germany) Avance DPX 300 spectrometer operating at 400 MHz. D2O was used as solvent. The chemical shifts are quoted relatively to tetramethylsilane standard.

#### 4.4.2. TEM

In order to characterize the nanofiller dispersion, transmission electron microscopy (TEM) was employed. Ultrathin slices (60 nm thick) of the dry nanocomposites were cut from each specimen under cryogenic conditions, using the Ultracut UTC ultramicrotome (from Leica, Wetzlar, Germany). The slices were subsequently put on supporting Cu grids and observed with the Tecnai G2 Spirit Twin 12 microscope (from FEI, Brno, Czech Republic) in the bright field mode at the acceleration voltage of 120 kV.

#### 4.4.3. Thermo-Mechanical Properties (DMTA)

Dynamic mechanical properties of the nanocomposite products were tested with rectangular platelet samples, using an ARES G2 apparatus from TA Instruments (New Castle, DE, USA—part of Waters, Milford, MA, USA). The analyzed temperature range was from −80 to +100 °C, the heating rate +3 °C min^−1^. An oscillatory shear deformation at the constant frequency of 1 Hz was applied, whose amplitude was varied between 0.01 and 5% (regulated by the ‘auto-strain’ function). The temperature dependences of the storage shear modulus (G′), of the loss modulus (G″) and of the loss factor tan(delta) were recorded. The standard specimen size was: 25 mm height, 10 mm width, and 0.4 mm thickness.

#### 4.4.4. DSC Analyses

The DSC analyses were carried out on a DSC822e instrument (from Mettler Toledo, Greifensee, Switzerland), using the STARe System software (Mettler Toledo). The DSC traces were recorded at the heating rate of 10 K/min, under a nitrogen flow of 60 mL/min. Calibration standards were indium and zinc, both supplied by Mettler Toledo.

#### 4.4.5. Simple Tensile Tests

Simple tensile properties of intact, as well as of self-healed samples were characterized using a universal testing machine Instron model 6025/5800R (from Instron Limited, High Wycombe, UK) equipped with a 100 N load cell, at room temperature, with a cross-head speed of 100 mm/min.

Only the thick specimens were used in this type of test. Such specimens were produced by ‘lamination’ (see above) of ‘film specimens’ obtained in the syntheses (by employing their self-healing properties) and were well suited for self-healing experiments, in contrast to the ‘film specimens’. Specimen dimensions (thick specimens): length: 30 mm, width: 5 mm, thickness: 4.5 mm.

At least five measurements were carried out for each tested material. Presented are the tensile curves closest to the average one.

#### 4.4.6. Cyclic Tensile Tests

The cyclic tensile tests were performed using an ARES-G2 (from TA Instruments, New Castle, DE, USA—part of Waters, Milford, MA, USA).

Standard cyclic characterizations were carried out as five repeated loading/unloading cycles. The loading part of each cycle was performed with the same relative crosshead speed (in %/min) as in the simple tensile tests, until a pre-defined maximum elongation, which was set equal to ca. 50% of the elongation at break of the given material. The unloading part of the cycle followed immediately, with the same crosshead speed.

In order to assess ‘internal self-healing’ (self-recovery), similar cyclic experiments were performed as above: they consisted of two above-described cycles separated by a time delay during which the sample was left to regenerate. In this way, the stretching/retraction cycle of the intact sample was recorded, and next the analogous cycle after a given regeneration time. For comparing results after several regeneration times, different ‘double cycles’ with different intact specimens as starting material had to be recorded. Standardly, ‘resting times’ of 30 and 60 min were applied (and compared with the result for the second cycle from the above-described 5-cycle test with no delay; hence the result from its second cycle corresponded to ‘regeneration time of 0 min’).

In the cyclic tests, always the ‘film’ samples were used. The standard geometry was: total length: 15 mm, width: 2 mm, thickness: 0.4 mm; the ‘active length’ between the clamps was 3–4 mm (not 15 mm).

## Data Availability

Data are contained within the article.

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
