# Peer review of "Self-Healing and Super-Elastomeric PolyMEA-co-SMA Nanocomposites Crosslinked by Clay Platelets"

_gels, 2022, doi:10.3390/gels8100657_

Round 1

Reviewer 1 Report

Authors deal interesting research regarding novel solvent-free ultra-extensible, tough, and self-healing nanocomposite elastomers. The influence of sodium methacrylate and clay filler on the properties of obtained composite elastomers was investigated. Paper is arranged well, logical and in proper manner. I suggest to accept the manuscript after few additional explanations:

-line 154, is reference 48 correct or reference 49 should be there?

-if authors have some images (optical, SEM or TEM images) of healed elastomers, especially healed region it will improve the quality and visual effect of the paper.

Author Response

Reviewer #1:

Authors deal interesting research regarding novel solvent-free ultra-extensible, tough, and self-healing nanocomposite elastomers. The influence of sodium methacrylate and clay filler on the properties of obtained composite elastomers was investigated. Paper is arranged well, logical and in proper manner. I suggest to accept the manuscript after few additional explanations:

Answer: The authors are very grateful for the positive assessment and for the valuable suggestions, which helped to improve the Manuscript and make it more attractive.

Reviewer #1:

-line 154, is reference 48 correct or reference 49 should be there?

Answer: The authors thank the Reviewer #1 for attentive reading. The suggestion is correct. The authors thoroughly checked the Literature list and found out, that because of format issues, the original reference 25 split into 25 and 26 in the submission file, and all the subsequent references were wrongly shifted by +1. The authors repaired this error in the literature list (its place is highlighted in red in the revised manuscript) and repaired and checked all the literature references. In the revised manuscript, the citation which is mentioned here became 48 again (but it is the content of the citation 49 in the original submission). Its position in the revised Manuscript is now on line 1055 if “Track Changes” is ON, or on line 863 if “Track Changes” is OFF (the chapter Experimental Part, now Materials and Methods namely was moved to the end of the Manuscript, as suggested by the Editor). The authors are very grateful to Reviewer #1 for discovering the hidden error in literature numbering.

Reviewer #1:

-if authors have some images (optical, SEM or TEM images) of healed elastomers, especially healed region it will improve the quality and visual effect of the paper.

Answer: This is indeed a very useful suggestion. The authors took a sharply focused and magnified optical photograph of the healed region, and added two detail images (from two directions) to the improved Figure 9, where self-healing tests are documented. In one direction of the detail image, the laminated layers of the thick sample are visible which were grown together (also using the self-healing property) to obtain the specimen, and which were perpendicular to the cut.

Reviewer 2 Report

In this work, the authors describe Super-Elastomeric PolyMEA-co-SMA Nanocomposites crosslinked by Clay Platelets. The study is well presented and written, and the results support the overall conclusion. However, how is this work different from previous work with nanoclays, laponite and so on?  See some papers below.

1.       https://www.astm.org/jte20190629.html

2.       https://www.sciencedirect.com/science/article/pii/S0032386109001451

3.       https://www.sciencedirect.com/science/article/pii/S2238785421012977

This reviewer does not consider this work novel enough to be publish in this journal.

Author Response

Reviewer #2:

In this work, the authors describe Super-Elastomeric PolyMEA-co-SMA Nanocomposites crosslinked by Clay Platelets. The study is well presented and written, and the results support the overall conclusion.

Answer: The authors are grateful for the positive overall assessment, for the attention, and for the useful discussion, which helped to improve the Manuscript and make it more attractive.

Reviewer #2:

However, how is this work different from previous work with nanoclays, laponite and so on?  See some papers below.

  1. https://www.astm.org/jte20190629.html
  2. https://www.sciencedirect.com/science/article/pii/S0032386109001451
  3. https://www.sciencedirect.com/science/article/pii/S2238785421012977

This reviewer does not consider this work novel enough to be publish in this journal.

Answer: The authors politely disagree with the above final conclusion. If the above-cited papers are considered:

-Paper 1, cited above by the Reviewer #2, is dedicated to asphalt binders containing several per cent nano-clay and SBS rubber as additives, and to the testing of their creep, elasticity and material fatigue. This interesting material is very different from the self-assembled and entangled brush-like structures investigated in the present work, not at least because of the dominant volume fraction of asphalt binder phase, and also because of the post-synthesis blending of SBS with nano-clay and binder.

-Paper 2, cited above, deals with super-elastomeric poly(N-isopropylacrylamide)/nano-clay hydrogels, doped by sodium methacrylate, which belong to the so-called Haraguchi-type hydrogels and which achieved ultra-extensibility combined with excellent recovery from very large deformations. However, in comparison with the studied solvent-free rubber-like super-elastomer, this material also is very different: The material from Paper 2 namely is a hydrogel, which contains a dominant amount of water (90 or more per cent). In comparison to this, the materials studied in the submitted work are solvent-free rubber-like elastomers, which are characterized by moduli orders of magnitude higher than in case of hydrogels, and which also are environmentally stable (they do not dry-up in air). In the rubber-like elastomers, super-extensibility is much more difficult to achieve than in hydrogels (as was mentioned in the original Introduction). This rare variety of super-elastomers was in the focus of interest of the presented manuscript. Nevertheless, for the readers of Gels, the studied solvent-free materials still can be fairly interesting, because their complex self-assembled structure is partly similar to the Haraguchi-type hydrogels (like in Paper 2) – the studied materials are solvent-free ‘relatives’ of the mentioned hydrogels. Some of them also are able of moderate or even strong swelling, although they lose their excellent mechanical properties in the swollen state.

-Paper 3 cited above is a previous paper by the authors dedicated to Haraguchi-type hydrogels doped with sodium methacrylate (related to the ones discussed in the above Paper 2), but the focus in Paper 3 was on generating the ability of rapid stimulus-response (T, pH) of large bulk non-porous gel specimens via stimulus-induced micro-phase separation. This type of nano-clay-filled hydrogel is different from the studied solvent-free elastomers in an analogous way, like the hydrogels in Paper 2.

-Generally, nano-clay indeed is a very popular nanofiller which was embedded in numerous matrixes. It can generate ‘simple’ nanofiller effects, as mentioned in the Introduction in the paragraph “Nanofillers”, or it can act via the ‘architecture effect’ (also mentioned in the “Nanofillers” paragraph) which means via self-assembly to complex architectures, as it is the case in Haraguchi-type gels, and also in the presently studied super-elastomers, and probably to some extent also in the above Paper 1.

 -A certain weakness of the original Introduction was, that after addressing several topics which led-up to the presented research work, it did not summarize, what is most original and novel in the newly presented materials. This was improved by adding the paragraph “In contrast to most of the above-discussed earlier works …”, just before “The aim of the present work”, where solvent-free super-elastomeric character (combining modulus typical of rubber with super-elasticity), the strong toughening-, self-healing- and self-recovery-supporting effect of the ionic dopant in this dry material, as well as the interesting effects of nano-phase-separation of the dopant in the hydrophobic matrix are highlighted. Abstract and Conclusions also were slightly improved in this respect. Also, the Figure 1 in the section “2.1.1. General properties, hydrophobicity vs. hydrophilicity” was improved, in order to highlight the solvent-free, rubbery nature of the studied nanocomposites.

 The authors are grateful for inspiring this improvement of the Manuscript.

Reviewer 3 Report

Dear Authors,

it was a pleasure to read your manuscript. However, some improvements are necessary.

At two places you write "force" followed by a mass. Please replace by pressure. Furthermore, the improvement of the chapter 2.2 is necessary. Your meaning is that reference 48 describes your experiments well. That is not true. In ref. 48 you describe experiments with 5 mL in contrast to round 50 mL in the manuscript. I expect a description which allows to reproduce your experiments. Eg. Nitrogen purging times are missed, the kind of pouring and the pouring volume are missed. The description of the kind of mixing must be improved. And here, please replace "weight of 2 kg" by the corresponding pressure.

Author Response

Reviewer #3:

Dear Authors,

it was a pleasure to read your manuscript. However, some improvements are necessary.

Answer: The authors are very grateful for the positive assessment and for the subsequent valuable suggestions, which helped to improve the Manuscript and make it more reader-friendly, especially in case of the description of the syntheses, and of samples processing.

Reviewer #3:

At two places you write "force" followed by a mass. Please replace by pressure.

Answer: The expression "force" followed by a mass (in the description of self-healing procedure in the “Materials and Methods” section, as well as in “Results and Discussion / Self-healing of disrupted samples and its efficiency” was replaced by pressure values (here 220 MPa). This indeed is far more descriptive for the situation in the sample during the healing process. The authors are very grateful for this wise suggestion.

Reviewer #3:

Furthermore, the improvement of the chapter 2.2 is necessary. Your meaning is that reference 48 describes your experiments well. That is not true. In ref. 48 you describe experiments with 5 mL in contrast to round 50 mL in the manuscript. I expect a description which allows to reproduce your experiments. Eg. Nitrogen purging times are missed, the kind of pouring and the pouring volume are missed. The description of the kind of mixing must be improved.

Answer: The chapter 2.2 (“Synthesis of the nanocomposite elastomers”, now 4.2) was improved very thoroughly, along the lines suggested, so that a reproduction should be now easy, and the text should be reader-friendly.

Concerning citation 48, the comments of Reviewer #3 and of Reviewer #1 helped to discover an error in the numbering of literature. Because of format issues, the original reference 25 split into 25 and 26 in the submission file, and all the subsequent references (including original 48) were wrongly shifted by +1. The authors repaired this error in the literature list (its place is highlighted in red in the revised manuscript) and repaired and checked all the literature references. In the revised manuscript, the citation which is mentioned here became 48 again (now correctly, but it is the content of the citation 49 in the original submission). The authors are very grateful to Reviewers #1 and #3 for discovering this hidden error. Independently from citation 48, the modified synthesis procedure used in the present work is now described in detail in the chapter “Synthesis of the nanocomposite elastomers” of the present manuscript.

Reviewer #3:

And here, please replace "weight of 2 kg" by the corresponding pressure.

Answer: In chapter 2.2 (“Synthesis of the nanocomposite elastomers”), the paragraph “Preparation of thick (‘laminated’) samples” was improved as suggested (the pressure here was 3 kPa). This is far more descriptive for the given situation. The authors thank for this useful suggestion.

Round 2

Reviewer 2 Report

The revised manuscript is suitable for publication. 

Thanks to the authors for answering the questions.